# Prevalence of Cardiovascular Comorbidities in Patients with Rheumatoid Arthritis

**DOI:** 10.3390/medicina60010038

**Published:** 2023-12-25

**Authors:** Marius Rus, Adriana Ioana Ardelean, Claudia Judea Pusta, Simina Crisan, Paula Marian, Liliana Oana Pobirci, Veronica Huplea, Alina Stanca Osiceanu, Gheorghe Adrian Osiceanu, Felicia Liana Andronie-Cioara, Madalina Ioana Guler

**Affiliations:** 1Department of Medical Disciplines, Faculty of Medicine and Pharmacy, University of Oradea, 410073 Oradea, Romania; paula.marian85@gmail.com; 2Faculty of Medicine and Pharmacy, University of Oradea, 410073 Oradea, Romania; adrianaardelean@uoradea.ro (A.I.A.); cjudeapusta@uoradea.ro (C.J.P.); opobirci@uoradea.ro (L.O.P.); hupleaveronica@uoradea.ro (V.H.); osiceanualina@yahoo.com (A.S.O.); adriana_ardelean@uoradea.ro (G.A.O.); fcioara@uoradea.ro (F.L.A.-C.); gulermadalina0106@gmail.com (M.I.G.); 3Department of Preclinical Disciplines, Faculty of Medicine and Pharmacy, University of Oradea, 410073 Oradea, Romania; 4Morphological Disciplines, Faculty of Medicine and Pharmacy, University of Oradea, 410073 Oradea, Romania; 5Cardiology Department, “Victor Babes” University of Medicine and Pharmacy, 2 Eftimie Murgu Sq., 300041 Timisoara, Romania; simina.crisan@umft.ro; 6Institute of Cardiovascular Diseases Timisoara, 13A Gheorghe Adam Street, 300310 Timisoara, Romania; 7Research Center of the Institute of Cardiovascular Diseases Timisoara, 13A Gheorghe Adam Street, 300310 Timisoara, Romania; 8Department of Psycho Neuroscience and Recovery, Faculty of Medicine and Pharmacy, University of Oradea, 410073 Oradea, Romania

**Keywords:** rheumatoid arthritis, cardiovascular complications, atherogenesis, cardiac arrhythmias, inflammation, disease, traditional factors, “non-traditional” factors

## Abstract

*Background and Objectives*: The risk of developing cardiovascular diseases (CVD) in patients suffering from rheumatoid arthritis (RA) is two times higher compared to the general population. The objective of this retrospective study was to determine which cardiovascular complications can appear in men vs. women with rheumatoid arthritis. Early diagnosis and initiation of therapeutic measures to reduce the progression rate of rheumatoid arthritis, while also maintaining an active lifestyle, are the most important problems in young patients. *Materials and Methods*: We included a number of 200 patients, divided into two groups according to gender (124 women and 76 men) with rheumatoid arthritis, presenting various stages of disease concomitant with cardiovascular complications. We assessed traditional and non-traditional risk factors, as well as electrocardiographic and echocardiographic findings in both groups. *Results*: All patients presented an atherogenic coefficient over two, indicating a significant risk of atherogenesis. Men had elevated levels of total cholesterol compared with women (≥200 mg/dL; 77.6%—men vs. 25.8%—women, *p* < 0.001). The participants presented cardiac arrhythmias, especially in the active stage of RA. Women had an increased risk of atrial fibrillation by 2.308 times compared to men (*p* = 0.020). One of the most important complications found in young women was pulmonary arterial hypertension (*p* = 0.007). *Conclusions*: In daily clinical practice, the screening of RA is carried out in sufficiently. This disease is often undiagnosed, and the risk factors remain unassessed. As a result, RA patients continue to present an increased risk of developing CVD.

## 1. Introduction

Rheumatoid arthritis (RA) is a systemic autoimmune disease defined by chronic inflammatory joint disease and extra-articular complications with a prevalence of 460 per 100,000 people [1]. Cardiovascular diseases (CVD) are common complications in rheumatoid arthritis patients, resulting in a more severe disease burden [2]. Patients with RA have an increased risk of developing coronary artery disease compared to the general population. RA patients are over three times more predisposed to suffer a myocardial infarction, while also presenting twice the risk of developing heart failure than healthy individuals. Some studies showed an increased risk of cerebrovascular disease, with the risk of stroke rising exponentially in RA patients, especially in younger individuals. In a Danish study, the incidence of atrial fibrillation (AF) in patients with RA was approximately 40% higher than in the general population [3]. Women had a higher risk of developing AF compared to men, and it was markedly increased in the youngest age groups. Compared to men, RA is more aggressive in women and the prognosis is worse. Sex hormones (especially estrogen) seem to play an important role in the regulation of the immune response [4].

The etiology of RA is based on the interaction between the patients’ genotypes and environmental factors. The risk of developing rheumatoid arthritis has been associated with HLA-DRB1 alleles. Several meta-analyses have indicated an increased risk of cardiovascular death by 50–60% [5,6]. This could be explained by the contribution of systemic inflammation in the development of atherogenesis, while the “traditional” cardiovascular risk factors are also attributed to nearly 50% of the total cardiovascular disease (CVD) risk [7].

Patients with active RA have significantly lower high- and low-density lipoprotein cholesterol (HDL-c/LDL-c) levels compared to the healthy population [8]. Statin treatment in patients with RA tends to reduce the levels of lipids and inflammatory factors. Smoking is also significantly associated with a higher Disease Activity Score, leading to worse clinical outcomes [9].

In the pathophysiology of RA, the risk of atherogenesis is increased by the inflammatory mechanisms involved in the disease. A useful marker of disease activity, elevated levels of C-reactive protein, hold a significant prognostic value [10]. The T-cell mediated cytotoxicity, characterizing the disease, as well as the circulating cytokines such as TNF- α, activates the endothelial cells, directly leading to endothelial injury and up-regulation of the adhesion molecules [11,12]. Even in the absence of identifiable CV risk factors, endothelial dysfunction is frequently present in RA patients and improves with anti-TNF-α therapy [13,14].

Numerous cardiac structures are involved in the pathological processes of RA, leading to atherosclerosis, arterial stiffness, coronary arteritis, congestive heart failure, valvular disease, and fibrinous pericarditis, thus increasing the risk of cardiovascular mortality [15]. Prognostic markers of diseases, such as hypertension and dyslipidemia, could be potentially included [16].

Speckle tracking assessment is useful for RA patients with unexplained dyspnea and a preserved ejection fraction. Impaired global longitudinal strain values, in combination with echocardiography findings of elevated filling pressures in the absence of valvular disease or pulmonary abnormalities, are consistent with heart failure with preserved ejection fraction. Also, speckle tracking is helpful for the assessment of the effects of anti-inflammatory treatment in RA.

The aim of this study was to determine the cardiovascular complications that can appear in men vs. women with rheumatoid arthritis, while also preventing their appearance through better control strategies of the “traditional” and “non-traditional” CV risk factors.

## 2. Materials and Methods

### 2.1. Study Design and Study Approval

In this retrospective study, 200 patients (124 women and 76 men) diagnosed with different stages of rheumatoid arthritis were included. All the patients were treated in the Cardiology Department of the Bihor County Clinical and Emergency Hospital, between March 2020 and March 2023. The study protocol was approved by the ethics committee of Bihor County Clinical and Emergency Hospital (decision no. 34/26 September 2019).

### 2.2. Inclusion and Exclusion Criteria

Figure 1 shows the inclusion and exclusion criteria used to describe the patients involved in this retrospective, observational and descriptive study.

### 2.3. Data Collection

The patients were divided into two groups:

Group I consisted of 124 women and group II included 76 men diagnosed with rheumatoid arthritis at various stages of the disease. We collected data regarding the main symptoms that have led to hospital admission, complete medical history, history of family diseases, and harmful behaviors. We looked for cardiovascular and non-cardiovascular risk factors. These rheumatoid arthritis patients presented cardiovascular risk factors such as smoking, alcohol consumption, hypertension, hypercholesterolemia, and diabetes. Non-cardiovascular risk factors included C-reactive protein, erythrocyte sedimentation rate, fibrinogen, and rheumatoid factor. We assessed the number of swollen and tender joints, most commonly affected in rheumatoid arthritis, based on the patient’s medical history. Additionally, we calculated DAS28 and CDAI scores and compared the scores. For each participant, we noted the ECG and echocardiographic findings, as well as specific medications used to treat rheumatoid arthritis.

Common cardiovascular diseases, observed in patients with rheumatoid arthritis, involved in this study, included accelerated atherosclerosis, coronary artery disease, arrhythmias such as atrial fibrillation, especially in women, pulmonary arterial hypertension, and pericarditis.

Coronary angiography was performed on some of the patients. The indication for this procedure was the presence of acute coronary syndrome. Before the intervention, the patients had been informed about the risks and the benefits of the procedure. The patients signed the informed consent of the procedure. The coronary angiography revealed monovascular, bivascular, and trivascular coronary artery disease.

The score Disease Activity Score 28 (DAS28) helps physicians assess the disease activity, based on the number of swollen and painful joints (the total of joints being 28) and acute-phase reactants, most commonly ESR, but CRP can be used instead. This score is calculated with the help of a computer, as it is not a simple sum. The scale of disease activity ranges from 0 to 10.
DAS28 = 0.56 × √(TJC) + 0.28 × √(SJC) + 0.36 × ln(CRP + 1) + 0.014 × PGA + 0.

TJC = Tender Joint Count; SJC = Swollen Joint Count; CRP = C-reactive protein; PGA = Patient Global Assessment of Disease Activity.

According to the DAS28 score, the disease is classified as follows:-Very active disease if DAS28 > 5.1;-Active if DAS is between 5.1 and 3.2;-Low disease activity if DAS is within 2.6–3.2;-Remission if DAS28 < 2.6 [17].

The clinical disease activity index (CDAI) is a simplification of the SDAI (Simplified Disease Activity Index). CDAI does not count for the value of acute phase reactants, using the same clinical assessments as the SDAI (Table 1).

CDAI is calculated as follows: SJC + TJC + PGA + EGA.

SJC = Swollen Joint Count; TJC = Tender Joint Count; PGA = Patient Global Assessment of Disease Activity; EGA = Evaluator Global Assessment of Disease Activity.

The SJC-28 joints are assessed, including shoulders, elbows, knees, and hands (first through the fifth metacarpophalangeal joint, the interphalangeal joint of the thumb, and the second through the fifth proximal interphalangeal joint);The TJC-28 joints are assessed, including shoulders, elbows, knees, and hands (first through fifth metacarpophalangeal joint, the interphalangeal joint of the thumb, and the second through fifth proximal interphalangeal joint).The PGA or Patient Global Disease Activity estimate represents the patient’s self-assessment of disease activity on a 0–10 scale where 10 represents maximal activity.The EGA or Evaluator Global Disease Activity estimate represents the evaluator’s assessment of disease activity on a 0–10 scale where 10 represents maximal activity [18].

The identification of rheumatoid arthritis was established based on:Medical history of joint symptoms (pain, tenderness, stiffness, swelling);Duration of joint symptoms (more than six weeks);Elevated levels of CRP or ESR;Positive biomarker test, like rheumatoid factor (RF);Radiographic changes of bones and joints.

Based on radiographic and clinical findings, rheumatoid arthritisis classified as follows:-Stage I: In the early stage of rheumatoid arthritis, the tissue around the joint(s) is inflamed. Pain and stiffness are common. The destructive changes in bones are not visible on X-rays.-Stage II: Moderate Stage RA—The inflammation has begun to damage the joint cartilage. Stiffness and a decreased range of motion can appear.-Stage III: Severe Stage RA—The severe inflammation causes bone damage. Higher pain, stiffness, and an even more decreased range of motions compared to stage 2 appear, as well as physical changes.-Stage IV: End-stage RA—In this stage, the inflammation stops, but the joint damage exacerbates. In this stage, severe pain, swelling, stiffness, and loss of joint mobility are the most common [19].

All patients received DMARDs, with 40 (20%) of them also taking NSAIDs. Remissive treatment was administered as follows: triple therapy (methotrexate, hydroxychloroquine, and sulfasalazine) to 50 patients (25%), DMARDs in monotherapy to 93 patients (46.5%), and DMARDs in dual therapy to 57 patients (28.5%). Corticosteroids were included in the remissive treatment of 86 patients (43%) (Table 2). None of the patients received biologic therapy because they refused to take this type of medication. All patients started statin treatment (atorvastatin 40 mg) upon admission to the Cardiology Department. The period of statin administration was 6 months. We evaluated the lipid profile before and after atorvastatin administration. Total cholesterol has a normal range lower than 200 mg/dL, LDL-cholesterol is normally between 70 to 130 mg/dL, HDL-cholesterol ranges from 40 to 60 mg/dL and triglycerides have a normal range of less than 150 mg/dL [20].

### 2.4. Statistical Analyses

All the data from the study was analyzed using IBM SPSS Statistics 25 and illustrated using Microsoft Office Excel/Word 2021. Quantitative variables were tested for normal distribution using the Shapiro–Wilk Test and were written as averages with standard deviations or medians with interquartile ranges. Quantitative independent variables with non-parametric distribution were tested between groups using Mann–Whitney *U* tests.

Quantitative paired variables with non-parametric distribution were tested between groups using the Related-sample Wilcoxon’s Signed Rank Test. Qualitative variables were written as counts or percentages and differences between groups were tested using Fisher’s Exact test/Pearson Chi-square Test. Z-tests with Bonferroni corrections were used to further detail the results obtained in the contingency tables.

Logistic regression univariable and multivariable models were used to predict the odds of cardiac complications and arterial fibrillation. The measure of prediction was quantified as odds ratios with 95% confidence intervals. The validity and significance of models were tested along with the goodness-of-fit tests and validation of the linearity assumption. In the case of atrial fibrillation, a multivariable model was constructed using the forward stepwise selection based on the Wald criteria.

## 3. Results

### 3.1. Patient Characteristics and Outcomes

The highest prevalence of the disease was observed in patients aged between 50 and 69 years old.

The majority of patients were diagnosed with stage III of the disease; 13% of patients were diagnosed during stage I, 19% of patients were diagnosed with stage II, 42.5% with stage III, and quite a high percentage of 25.5% were diagnosed with stage IV (Table 3).

According to the data presented in Table 3, men were significantly more associated with the dyslipidemia criteria, having more frequently elevated levels of total cholesterol (≥200 mg/dL) (77.6% men vs. 25.8% women, Fisher’s Exact Test—*p* < 0.001), LDL-cholesterol (≥140 mg/dL) (72.4% men vs. 22.6% women, Fisher’s Exact Test—*p* < 0.001), triglycerides (≥150 mg/dL) (81.6% men vs. 25.8% women, Fisher’s Exact Test—*p* < 0.001) and lower concentrations of HDL-cholesterol (<40 mg/dL) (25% men vs. 8.1% women, Fisher’s Exact Test—*p* = 0.002).

There was a higher incidence in men regarding traditional and non-traditional CV risk factors. CRP, ESR, fibrinogen, and rheumatoid factor values were increased in men compared to women (Table 3).

As displayed in the table above, men presented a higher incidence of coronary disease compared to women (Fisher’s Exact Test with Bonferroni corrected Z-tests—*p* < 0.001). An amount of 42.1% of men presented monovascular coronary artery disease, 19.7% bivascular coronary artery disease, and 11.8% trivascular coronary artery disease. Women exhibited monovascular coronary artery disease in 24.1% of cases, bivascular coronary artery disease in 9.6% of cases, and trivascular coronary artery disease in 4% of cases (Table 3).

Electrocardiogram examinations were recorded for every patient enrolled in this study. Left ventricular hypertrophy was significantly more frequent in men (51.3%) than women (27.4%), *p* = 0.001. Secondary ST-T appeared in 39.5% of men and 26.6% of women (showing a tendency towards statistical significance in the direction of higher frequencies in men than women, *p* = 0.062), while atrial fibrillation was more frequently in women (32.3%) than men (17.1%) (*p* = 0.021). Thus, a higher incidence of ECG abnormalities could be observed in men. The examination of the ECGs highlighted the presence of arrhythmias and conduction disorders (supraventricular extrasystoles, left anterior fascicular block, left bundle branch block, premature ventricular contractions, and atrial fibrillation) (Table 3).

The echocardiography confirmed the presence of ventricular hypertrophy, previously reported on the ECGs. Moreover, pericarditis, pulmonary arterial hypertension, and mitral valve prolapse appeared to have a higher incidence in women compared to men, although significant differences between genders were not observed (Fisher’s Exact Tests—pericarditis (*p* = 0.174), mitral valve prolapse (*p* = 0.083), mitral regurgitation (*p* = 0.771), pulmonary arterial hypertension (*p* = 0.199), tricuspid regurgitation (*p* = 0.874), and pulmonary regurgitation (*p* = 0.880)), while wall motion abnormalities (hypokinesia and akinesia) were present in 31 patients (17 men and 14 women), possibly due to past myocardial infarctions. Young participants, particularly women, had a high incidence of pulmonary arterial hypertension (Table 3).

The main cause of heart failure was arterial hypertension (44.5%), while coronary artery disease was responsible for 17.5%, valvular heart disease for 14% of cases, and arrhythmias and conduction disorders for 7.5% of cases (Table 3).

Data from Table 4 show the comparison of lipid profile parameters in evolution before and after atorvastatin treatment. The distribution of all variables was non-parametric for all measurements according to the Shapiro–Wilk test (*p* < 0.001). The differences between pre- and post-atorvastatin treatment for all lipid profile parameters were statistically significant according to the Wilcoxon tests, showing the following:-A decrease in total cholesterol (pre: median = 172.5, IQR = 155–229 vs. post: median = 150, IQR = 145–200) (*p* < 0.001) the observed difference being significant (23.78 ± 18.9, median = 20 (IQR = 10–30.75));-A decrease in triglycerides (pre: median = 120, IQR = 95–160 vs. post: median = 100, IQR = 90–150) (*p* < 0.001) the observed difference being significant (14.32 ± 22.95, median = 10 (IQR = 5–25));-A decrease in LDL-cholesterol (pre: median = 110, IQR = 92.25–153 vs. post: median = 90.5, IQR = 82.25–129 (*p* < 0.001) the observed difference being significant (22.34 ± 17, median = 18 (IQR = 11–30.75));-An increase in HDL cholesterol (pre: median = 40, IQR = 40–44 vs. post: median = 42, IQR = 41–45) (*p* < 0.001) the observed difference being significant (1.13 ± 2.5, median = 1 (IQR = 0–2)).

As we can see in Table 5, women had a higher incidence regarding the damage of joints in RA compared to men.

Data from Table 6 presents DAS28 and CDAI scores in both groups. Women presented a higher incidence of the DAS score between 5.1 and 3.2 (active disease) compared to men, who had a higher incidence of the DAS score between 3.2 and 2.6 (low disease activity). Regarding the CDAI score, women had a higher incidence of moderate activity disease index (10.1–22.0) compared to men who had a higher incidence of low activity disease index (2.9–10.0).

Data from Table 7 and Figure 2 show the correlation between CDAI and DAS28. The distribution of both variables was non-parametric according to the Shapiro–Wilk test (*p* < 0.001). The correlation observed was statistically significant (*p* = 0.022, R = −0.162), being a weak negative correlation, which shows that patients with high values of the CDAI score were significantly more associated with lower values of DAS28 scores and vice versa.

As displayed in Table 8, 70% of patients (140 individuals) presented an ESR exceeding the upper limits, while the remaining cases presented values within the normal range. Additionally, C-reactive protein levels were within the normal range for 34.5% of patients (69 cases), while 65.5% (132 patients) displayed elevated C-reactive protein values.

Based on collected data, we determined the atherogenic coefficient, with men scoring an average value of 4.52 +/− 0.27 (median = 4.60, IQR = 4–5) and women scoring 3.13 +/− 0.20 (median = 2.75, IQR = 2.37–4.10), the Mann–Whitney U Test showing higher values in men versus women (*p* < 0.001). In conclusion, both groups scored an atherogenic coefficient over 2, indicating a higher risk of atherogenesis, more frequently in men (100%) than women (93.5%) (Fisher’s Exact Test—*p* = 0.025) (Figure 3).

### 3.2. Correlation of Patients’ Parameters

The comparison of the analyzed parameters between patients according to the existence of cardiac complications showed that in patients with pericarditis, only HDL-cholesterol and atrial fibrillation were significantly different between groups. Patients with pericarditis had significantly higher values of HDL-cholesterol (median = 42, IQR = 40–45) compared to patients without pericarditis (median = 40, IQR = 40–43) (*p* = 0.039). Also, atrial fibrillation was significantly more frequently associated with pericarditis (41.2% vs. 23.5%, (*p* = 0.033).

Data from Table 9 show the logistic regression models used for the prediction of cardiac complications in patients with mitral valve prolapse. Only the frequency of LVH was significantly different between groups, as patients with LVH were significantly less associated with mitral valve prolapse (40.9% vs. 21.7%) than patients without LVH (78.3% vs. 59.1%) (*p* = 0.023). CRP, triglycerides, HDL-cholesterol, and secondary ST-T changes were significantly different between groups in patients with mitral regurgitation. These patients had higher CRP values (median = 121, IQR = 4–150 vs. median = 105, IQR = 2–132.5, *p* = 0.012), lower values of HDL-cholesterol (median = 40, IQR = 40–42 vs. median = 42, IQR = 40–44, *p* = 0.018), higher frequencies of triglycerides dyslipidemia (55.2% vs. 40.7%, *p* = 0.047), and lower frequencies of secondary ST-T changes (38.1% vs. 23%, *p* = 0.031) than patients without mitral regurgitation.

In the case of pulmonary arterial hypertension, only rheumatoid factor and HDL-cholesterol were significantly different between groups, patients with pulmonary arterial hypertension presenting significantly higher values of rheumatoid factor (median = 206, IQR = 137–216 vs. median = 202, IQR = 123–211, *p* = 0.036) and higher values of HDL-cholesterol (median = 42, IQR = 40–45 vs. median = 40, IQR = 40–43, *p* = 0.045) in comparison to patients without pulmonary arterial hypertension. In the case of tricuspid regurgitation, none of the parameters were significantly different between groups (*p* > 0.05).

Left ventricular hypertrophy was a significant predictor for mitral valve prolapse (*p* = 0.020). Patients without left ventricular hypertrophy present increased odds of developing mitral valve prolapse by 2.493 times (95% C.I.: 1.153–5.376). In the case of mitral regurgitation, in univariate models, HDL-cholesterol was not a significant predictor (*p* = 0.055), while CRP (*p* = 0.019), triglycerides dyslipidemia (*p* = 0.043) and secondary ST-T changes (*p* = 0.024) were significant predictors. Each increase of 1 unit of CRP was associated with increased odds of having mitral regurgitation by 1.005 times (95% C.I.: 1.001–1.009). Patients with dyslipidemia presented increased odds of having mitral regurgitation by 1.793 times (95% C.I.: 1.019–3.154). Patients without secondary ST-T changes presented increased odds of having mitral regurgitation by 2.057 times (95% C.I.: 1.098–3.861).

In the case of pulmonary arterial hypertension, none of the variables were significant predictors (*p* > 0.05). For pulmonary regurgitation, the left bundle branch block was a significant predictor (*p* = 0.009), as patients with this condition had increased odds of developing pulmonary regurgitation by 7.264 times (95% C.I.: 1.656–31.869).

Data from Table 10 show the comparison of analyzed parameters between patients according to the existence of atrial fibrillation. Men were significantly less associated with atrial fibrillation (42.9% vs. 24.5%) than women (75.5% vs. 57.1%) (*p* = 0.021). Patients with arterial hypertension were significantly less associated with atrial fibrillation (57.8% vs. 39.6%) than patients without arterial hypertension (60.4% vs. 42.2%) (*p* = 0.025). Fibrinogen levels were significantly lower in patients with atrial fibrillation (median = 4.6, IQR = 4.5–4.85) than in patients without atrial fibrillation (median = 4.7, IQR = 4.5–5.1) (*p* = 0.022). Rheumatoid factor levels were significantly higher in patients with atrial fibrillation (median = 206, IQR = 168.5–213.5) than in patients without atrial fibrillation (median = 201, IQR = 121–213) (*p* = 0.047).

Patients without coronary artery disease were significantly more associated with atrial fibrillation (62.3% vs. 43.5%) (*p* = 0.030).

Data from Table 11 shows the logistic regression models used for the prediction of atrial fibrillation. In univariate models, each of the analyzed parameters were significant predictors (*p* < 0.05). Female patients had increased odds of atrial fibrillation by 2.308 times (95% C.I.: 1.139–4.674) (*p* = 0.020). Non-smokers had increased odds of atrial fibrillation by 2.557 times (95% C.I.: 1.219–5.376) (*p* = 0.013) and non-hypertensive patients had increased odds of atrial fibrillation by 2.087 times (95% C.I.: 1.101–3.968) (*p* = 0.024). Each decrease of 1 unit of fibrinogen had increased odds of atrial fibrillation by 3.322 times (95% C.I.: 1.183–9.345) (*p* = 0.023), each increase of 1 unit of rheumatoid factor had increased odds of atrial fibrillation by 1.009 times (95% C.I.: 1.001–1.017) (*p* = 0.020). Also, patients without coronary artery disease had increased odds of atrial fibrillation by 1.972 times (95% C.I.: 1.008–3.861) (*p* = 0.047). Each decrease of 1 unit of atherogenic coefficient had increased odds of atrial fibrillation by 1.388 times (95% C.I.: 1.055–1.828) (*p* = 0.019) and each decrease of 1 unit of total cholesterol had increased odds of atrial fibrillation by 1.010 times (95% C.I.: 1.002–1.018) (*p* = 0.011).

The multivariable model was selected using the forward stepwise selection. Total cholesterol, coronary artery disease, and pericarditis were the selected variables in the prediction, all of them being independent significant predictors.

## 4. Discussion

This study aimed to identify cardiovascular complications in men vs. women with rheumatoid arthritis (RA), with the purpose of reducing cardiovascular risk through comprehensive management of traditional and non-traditional risk factors. In this retrospective study, we enrolled 200 patients; the majority of them were women, with the highest prevalence of the disease observed in patients aged between 50 to 69 years old. A meta-analysis of 14 controlled observational studies involving 41,490 patients reported that the risk of CVD increases by over 48% in RA patients compared to the general population [21].

Rheumatoid arthritis is two- to three times more frequent in women than in men, and a strong association with sex hormones has been demonstrated. There is strong evidence that autoimmunity is under genetic control, and genes in sexual chromosomes can play a role in supporting female prevalence. On the other hand, it is widely accepted that sex hormones—estrogens in particular—may regulate the immune response by promoting the survival of forbidden autoreactive clones and ultimately the prevalence of autoimmunity in women. Accordingly, estrogens have been suggested to be associated with the development of RA [4]. As you can see, our study and other studies show that women are at a higher risk of developing RA compared to men, the reason being the important role the sex hormones (estrogen) play in autoimmunity.

It was interesting to observe that there were several cases of young RA patients, particularly females, who had cardiac complications, such as pulmonary arterial hypertension, which suggests a potential connection between early onset of RA and premature cardiovascular comorbidities. Pulmonary arterial hypertension may be associated with RA, due to chronic systemic inflammation, especially in the respiratory system, which can lead to endothelial cell injury, inflammation, coagulation, vasoconstriction, and fibrosis. A particularly grim prognosis with faster progression of disease and poor response to therapy is found in patients with pulmonary arterial hypertension (PAH) associated with RA [22]. Additionally, a case–control study assessed potential cardiac abnormalities in 47 RA patients without manifested cardiovascular symptoms, using the Doppler echocardiography technique, emphasizing a high incidence of pulmonary hypertension and left ventricular diastolic dysfunction [22].

Both groups exhibited elevated atherogenic coefficients, indicating an increased atherogenesis risk. Men also presented elevated levels of LDL-cholesterol and an increased prevalence of smoking than women. Atherosclerotic plaques, the main CVD mechanism in RA, originate from the interaction between vascular, metabolic, and inflammatory factors. Dyslipidemia, hypertension, smoking, and inflammation contribute to arterial endothelial damage, thus causing atherosclerosis. The plaques may grow gradually or become unstable, leading to acute cardiovascular events [23]. The JAK/STAT signaling pathway, regulated by cytokines like interleukin-6, plays a role in RA pathogenesis, increasing systemic inflammation and extra-articular comorbidities, such as atherosclerosis [8,9,24]. Several studies have shown that chronic inflammation, in combination with dyslipidemia and smoking, plays an important role in the development of atherosclerotic plaques.

The European Society of Cardiology (ESC) recognizes the heightened risk of RA but provides few specific recommendations. The guidelines suggest a thorough evaluation of total CVD risk in adult patients with a lower threshold and consider multiplying the calculated risk by 1.5 based on disease activity [25]. Additionally, for the management of CVD risk, the ESC recommends similar interventions to the general high-risk population.

The differences between pre- and post-atorvastatin treatment for both groups were statistically significant (*p* < 0.001). Among statin-treated patients, an improved lipid profileand reduced inflammatory markers (ESR, CRP) were observed, especially in men. A meta-analysis on the use of statins in patients with RA indicated that they may have an anti-inflammatory effect, as well as a lipid-lowering property [26,27], with atorvastatin reducing the Disease Activity Score more than simvastatin. However, there are conflicting reports where statins are associated with an increased risk of developing RA in the first year of use [28].

ECG findings included left ventricular hypertrophy, secondary ST-T changes, left bundle branch block, atrial fibrillation, premature ventricular contraction, sinus tachycardia, and supraventricular extrasystoles, particularly during the active periods of the disease. The incidence of atrial fibrillation was higher in women. The systemic inflammatory state in chronic conditions such as RA may increase the risk of AF. Studies suggest a significantly increased incidence of AF in RA patients compared to the general population [13], with a Danish nationwide cohort study by Lindhardsen et al. reporting an overall 40% higher incidence [3]. Using data from a large US commercial insurance plan, Kim et al. found that hospitalization for AF in RA patients was 1.4 times higher than in non-RA patients [29].

Comparable to our results, Shenavar-Masooleh et al. [30] focused on a study of 100 RA patients and found that 32% presented abnormal ECG findings. The most common findings were ST-segment and T-wave changes, similar to those reported in our study, occurring in 46.9% of cases. Moreover, sinus tachycardia and low voltage were both observed in 3.1% of cases, premature ventricular tachycardia in 6.2%, axis deviation and poor R progression in 9.3%, branch block in 12.5%, and pulmonary *p* in 15.6% of cases.

The echocardiography assessment revealed pericarditis, pulmonary arterial hypertension, mitral valve prolapse, mitral and tricuspid regurgitation, and pulmonary regurgitation. We observed no significant differences between genders regarding cardiac valvulopathies. Mac Donald et al. [31] achieved clinical, electrocardiographic (ECG), and echocardiography examinations in 51 American outpatients with RA. Thirty-one percent of patients presented echocardiographic evidence of pericardial effusion, with pericardial thickening demonstrated in two patients. The authors concluded that in unselected outpatients with RA, pericardial abnormalities detected by echocardiography are common, although usually clinically unapparent. Pericardial effusion is one of the most frequent echocardiographic findings in RA patients. In our study, only 20% of women and 12% of men presented minimum pericarditis as an inflammatory reaction. Toumanidis et al. [32] observed mitral and aortic cusps changes in about 24% of their patients. Our patients presented minor mitral valve prolapse and minor mitral regurgitation, especially in women.

Common causes of congestive heart failure include coronary artery disease and arterial hypertension. Patients with RA have approximately double the risk of atherosclerotic CVD, stroke, heart failure, and atrial fibrillation (AF) compared to the general population [33,34,35]. Furthermore, patients with active RA develop more CV events and an increased mortality risk.

The exact etiology of RA remains incompletely understood. However, both genetic and environmental factors contribute to its development. Several genetic and epigenetic components have been linked to RA, in addition to various environmental factors like cigarette smoke, dust exposure, and the microbiome. Disease progression often begins years prior to symptom onset, with the development of specific autoantibodies, including rheumatoid factor (RF) and anti-citrullinated protein antibodies (ACPA). ACPA-positive RA patients typically experience more severe disease activity and an elevated risk of cardiovascular mortality. The presence of these antibodies may also contribute to the atherosclerotic process. Interestingly, ACPA has been identified in non-RA patients with cardiovascular disease, correlating with worse cardiovascular outcomes [36].

According to numerous studies, physical activity significantly reduces inflammation, relieves arthritis pain, and improves mobility, making it a recommended treatment by the American College of Rheumatology [37]. The pattern of the disease is fluctuant, with episodic exacerbations and remission periods. Without optimal treatment, symptoms worsen over time, with the joints gradually becoming irreversibly damaged [38]. Life expectancy is reduced by several years, due to RA complications and comorbidities [38]. The target of treatment is to reduce any side effects and go into remission [39]. Pharmacological agents that help maintain joint function can be classified as conventional synthetic disease-modifying antirheumatic drugs (DMARDs), biologic DMARDs, and targeted synthetic DMARDs, which are included in a new class of nonbiologic DMARDs by the American College of Rheumatology (ACR) [40]. Nonsteroidal anti-inflammatory drugs (NSAIDs) and glucocorticoids (GCs) are often used as adjunctive therapy in reducing inflammation when patients present inadequate symptom control [41].

Glucocorticoids, like prednisolone, may elevate cardiovascular risk among RA patients in a dose- and duration-dependent manner. In an extensive retrospective registry study by Ocon et al., RA patients who used below five milligrams daily, or had a cumulative dose lower than 750 mg in total, did not experience an increased CV risk [41]. However, patients with higher daily doses, greater cumulative doses, or longer treatment durations, did face an elevated CV risk. In the recent GLORIA trial, where patients received low-dose glucocorticoids (5 mg prednisolone) for 24 months, a 35% increase in the incidence of CV events was found (2.4 cases per 100 patient years in the prednisolone group vs. 1.7 in the placebo group) [10]. Selective COX-2 inhibitors are known to increase the CV risk in the general population by approximately 35–40% [12]. This increase is comparable to that of traditional NSAIDs such as ibuprofen or diclofenac [14].

The strengths of this retrospective study were the facile means of accessing the patient database, no costs for doing the study, and the age diversity of patients (inclusion criteria being the wide age range of between 18 and 90 years old).

The study limitations were as follows:-The study sample is relatively small (200 participants). A larger and more diverse sample could provide a more accurate representation of the cardiovascular complications of these patients;-Some data are not reported by participants;-The duration of this study (3 years) might not capture all cardiovascular complications in young patients with rheumatoid arthritis;-Possible selection bias.

We are conscious that our research may have limitations. For example, the number of RA patients is relatively small. However, many studies had a similar number of patients. Further, we did not have clinical endpoints and longitudinal follow-up, and therefore, we were not able to assess the clinical impact of our findings.

## 5. Conclusions

The results of this study suggest that patients with rheumatoid arthritis present an increased risk of cardiovascular complications. Men presented a higher incidence of traditional cardiovascular and non-traditional risk factors compared to women. Regular cardiovascular risk assessment is important in patients with rheumatoid arthritis. Traditional risk factors are often underdiagnosed, particularly in young patients, leading to increased cardiovascular morbidity and mortality.

One of the most important complications in young RA patients included in our study was pulmonary arterial hypertension. If untreated, this disease can lead to right-sided heart failure and increased mortality. Also, the examination of the ECGs highlighted a higher incidence of arrhythmias and conduction disorders in men, except for atrial fibrillation found more frequently in women.

Prevention strategies and comprehensive therapeutic schemes require close collaboration between the rheumatologist, cardiologist, and primary care physician of each RA patient. The severity of the disease, the cardiovascular risk factors, as well as the age of each affected individual, must be considered. Patients with RA should be advised of possible cardiovascular complications and their early manifestations.

## Figures and Tables

**Figure 1 medicina-60-00038-f001:**
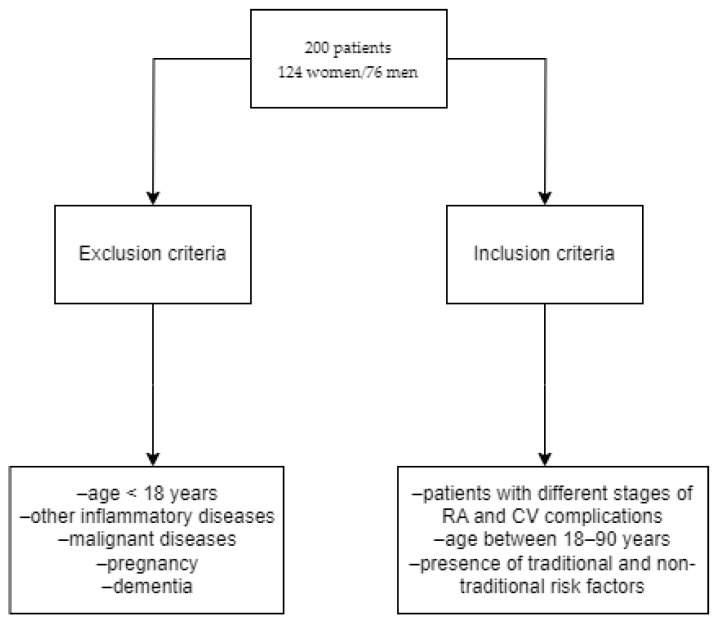
Inclusion and exclusion criteria.

**Figure 2 medicina-60-00038-f002:**
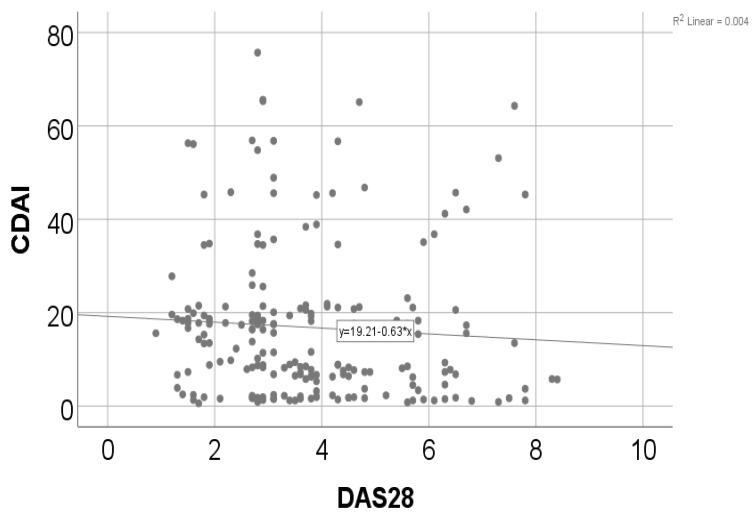
Correlation between CDAI and DAS28.

**Figure 3 medicina-60-00038-f003:**
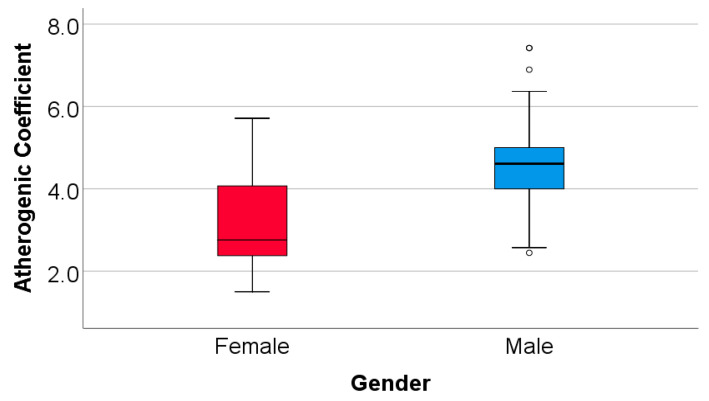
Comparison of atherogenic coefficient according to gender.

**Table 1 medicina-60-00038-t001:** CDAI score interpretation.

Clinical Disease Activity Index (CDAI)	
0.0–2.8	Remission
2.9–10.0	Low activity
10.1–22.0	Moderate activity
22.1–76.0	High activity

**Table 2 medicina-60-00038-t002:** Distribution of patients according to treatment.

Therapy	Number of Cases	Percentage (%)
DMARDs	200	100
DMARDs + NSAIDs	40	20
DMARDs in monotherapy	93	46.5
DMARDs in dual therapy	57	28.5
MTX + HCQ + SSZ	50	25
Corticosteroids	86	43

DMARDs = disease-modifying antirheumatic drugs; NSAIDs = nonsteroidal anti-inflammatory drugs; MTX = methotrexate; HCQ = hydroxychloroquine; SSZ = sulfasalazine.

**Table 3 medicina-60-00038-t003:** Analyzed characteristics in the entire study group and according to gender.

Parameter	Total	Men (*N* = 76)	Women (*N* = 124)	*p*
Age group (Nr., %)				
40–49 years	25 (12.5%)	11 (14.5%)	14 (11.3%)	0.291 *
50–69 years	112 (56%)	46 (60.5%)	66 (53.2%)	
≥70 years	63 (31.5%)	19 (25%)	44 (35.5%)	
Stage of RA (Nr., %)				
Stage I	26 (13%)	13 (17.1%)	13 (10.5%)	0.004 *
Stage II	38 (19%)	17 (22.4%)	21 (16.9%)	
Stage III	85 (42.5%)	37 (48.7%)	48 (38.7%)	
Stage IV ^†^	51 (25.5%)	9 (11.8%)	42 (33.9%)	
Smoking (Nr., %)	70 (35%)	50 (65.8%)	20 (16.1%)	<0.001 *
Consumption of alcohol (Nr., %)	47 (23.5%)	32 (42.1%)	15 (12.1%)	<0.001 *
Hypertension (Nr., %)	106 (53%)	68 (89.5%)	38 (30.6%)	<0.001 *
Hypercholesterolemia (Nr., %)	91 (45.5%)	59 (77.6%)	32 (25.8%)	<0.001 *
Diabetes mellitus (Nr., %)	28 (14%)	15 (19.7%)	13 (10.5%)	0.054 *
CRP (mg/L) (Mean ± SD, Median (IQR))	89.04 ± 67.6, 110 (3–142)	94.68 ± 73.7, 123 (2–153.75)	79.83 ± 55.5, 105.5 (3–116.5)	0.006 **
Fibrinogen (g/L) (Mean ± SD, Median (IQR))	4.77 ± 0.32, 4.7 (4.5–5.1)	5.14 ± 0.15, 5.2 (5.1–5.2)	4.54 ± 0.14, 4.6 (4.5–4.6)	<0.001 **
Rheumatoid factor (UI) (Mean ± SD, Median (IQR))	177.52 ± 45.86, 203.5 (125–213)	212.22 ± 10.6, 210 (205–216.75)	120.91 ± 12.61, 119 (111–132)	<0.001 **
Atherogenic coefficient (Mean ± SD, Median (IQR))	3.655 ± 1.251, 3.25 (2.571–4.731)	4.518 ± 1.08, 4.61 (4–5)	3.127 ± 1.04 2.75 (2.37–4.1)	<0.001 **
ESR (mm/h) (Mean ± SD, Median (IQR))	76.98 ± 44.94, 98 (15–106.75)	75.21 ± 43.46, 97.5 (15–105)	78.06 ± 45.97, 98.5 (15–107)	0.552 **
Dyslipidemia—Total cholesterol (Nr., %)	91 (45.5%)	59 (77.6%)	32 (25.8%)	<0.001 *
Dyslipidemia—Tryglicerides (Nr., %)	94 (47%)	62 (81.6%)	32 (25.8%)	<0.001 *
Dyslipidemia—HDL-cholesterol (Nr., %)	29 (14.5%)	19 (25%)	10 (8.1%)	0.002 *
Dyslipidemia—LDL-cholesterol (Nr., %)	83 (41.5%)	55 (72.4%)	28 (22.6%)	<0.001 *
Coronary artery disease (Nr., %)				
Absent	97 (48.5%)	20 (26.3%)	77 (62.1%)	<0.001 *
Monovascular coronary a. disease	62 (31%)	32 (42.1%)	30 (24.2%)	
Bivascular coronary a. disease	27 (13.5%)	15 (19.7%)	12 (9.7%)	
Trivascular coronary a. disease	14 (7%)	9 (11.8%)	5 (4%)	
Sinus tachycardia (Nr., %)	40 (20%)	15 (19.7%)	25 (20.2%)	1.000 *
Left ventricular hypertrophy (Nr., %)	73 (36.5%)	39 (51.3%)	34 (27.4%)	0.001 *
Secondary ST-T changes (Nr., %)	63 (31.5%)	30 (39.5%)	33 (26.6%)	0.062 *
Atrial fibrillation (Nr., %)	53 (26.5%)	13 (17.1%)	40 (32.3%)	0.021 *
Supraventricular extrasystole (Nr., %)	42 (21%)	20 (26.3%)	22 (17.7%)	0.210 *
Left anterior fascicular block (Nr., %)	32 (16%)	14 (18.4%)	18 (14.5%)	0.552 *
Premature ventricular contraction (Nr., %)	26 (13%)	12 (15.8%)	14 (11.3%)	0.391 *
Left bundle branch block (Nr., %)	24 (12%)	10 (13.2%)	14 (11.3%)	0.823 *
Pericarditis (Nr., %)	34 (17%)	9 (11.8%)	25 (20.2%)	0.174 *
Mitral valve prolapse grade I-II (Nr., %)	46 (23%)	12 (15.8%)	34 (27.4%)	0.083 *
Mitral regurgitation grade I-II (Nr., %)	87 (43.5%)	32 (44.1%)	55 (44.4%)	0.771 *
Pulmonary arterial hypertension (Nr., %)	39 (19.5%)	11 (14.5%)	28 (22.6%)	0.199 *
Tricuspid regurgitation grade I-II (Nr., %)	139 (69.5%)	52 (68.4%)	87 (70.2%)	0.874 *
Pulmonary regurgitation grade I-II (Nr., %)	128 (64%)	48 (63.2%)	80 (64.5%)	0.880 *
Etiology—CHF—High blood pressure (Nr., %)	89 (44.5%)	38 (50%)	51 (41.1%)	0.243 *
Etiology—CHF—Coronary artery disease (Nr., %)	35 (17.5%)	15 (19.7%)	20 (16.1%)	0.567 *
Etiology—CHF—Valvular heart disease (Nr., %)	28 (14%)	12 (15.8%)	16 (12.9%)	0.675 *
Etiology—CHF—Arrhythmias and conduction disorders (Nr., %)	15 (7.5%)	5 (6.6%)	10 (8.1%)	0.788 *

* Fisher’s Exact Test, ** Mann–Whitney *U* Test, ^†^ Stage IV of RA disease significantly more frequent in women than men (according to Bonferroni corrected Z-tests), CHF = congestive heart failure, IQR = interquartile range, SD = standard deviation, RA = rheumatoid arthritis.

**Table 4 medicina-60-00038-t004:** Comparison of lipid profile parameters in evolution before and after atorvastatin treatment * Related-samples Wilcoxon Signed Rank Test.

Parameter/Measurement		Pre-Atorvastatin	Post-Atorvastatin	*p* *
Total cholesterol	Average ± SD	191 ± 43.33	167.22 ± 32.83	<0.001
	Median (IQR)	172.5 (155–229)	150 (145–200)	
Triglycerides	Average ± SD	128.68 ± 39.02	114.35 ± 31.45	<0.001
	Median (IQR)	120 (95–160)	100 (90–150)	
HDL-cholesterol	Average ± SD	41.54 ± 2.61	42.67 ± 2.81	<0.001
	Median (IQR)	40 (40–44)	42 (41–45)	
LDL-cholesterol	Average ± SD	123.71 ± 38.2	101.37 ± 28.61	<0.001
	Median (IQR)	110 (92.25–153)	90.5 (82.25–129)	

**Table 5 medicina-60-00038-t005:** Distribution of the patients depending on the joint involvement in RA, MCP, metacarpophalangeal joints; PIP, proximal interphalangeal joints; DIP, distal interphalangeal joints; CMC, carpometacarpal joints.

Joint Involvement in RA	Women	Men
Wrists	64 (51.6%)	33 (43.4%)
MCP	67 (54%)	32 (42.1%)
PIP	54 (43.5%)	15 (19.7%)
Spares DIP	44 (35.4%)	22 (28.9%)
First CMC	30 (24.1%)	12 (15.7%)

**Table 6 medicina-60-00038-t006:** DAS28 and CDAI score according to gender; DAS28 = Disease Activity Score and 28 refers to the 28 joints that are assessed; CDAI = Clinical Disease Activity Index.

**DAS28**	**>5.1**	**5.1–3.2**	**3.2–2.6**	**<2.6**
Men	15 (19.73%)	20 (26.31%)	24 (31.57%)	17 (22.36%)
Women	28 (22.58%)	44 (35.48%)	27 (21.77%)	25 (20.16%)
**CDAI score**	**0.0–2.8**	**2.9–10.0**	**10.1–22.0**	**22.1–76.0**
Men	17 (22.36%)	23 (30.26%)	21 (27.63%)	15 (19.73%)
Women	26 (20.96%)	28 (22.58%)	46 (37.09%)	24 (19.35%)

**Table 7 medicina-60-00038-t007:** Correlation between CDAI and DAS28. * Spearman’s rho Correlation Coefficient, ** Shapiro–Wilk Test.

Correlation	*p* *
CDAI (*p* < 0.001 **) × DAS28 (*p* < 0.001 **)	0.022, R = −0.162

**Table 8 medicina-60-00038-t008:** Biomarkers of inflammation in patients with rheumatoid arthritis.

Inflammation Biomarkers	Number of Cases	Percentage (%)
ESR	140	70
CRP	131	65.5

**Table 9 medicina-60-00038-t009:** Logistic regression models used for the prediction of cardiac complications.

Pericarditis				
Parameter	Univariable		Multivariable	
	OR (95% C.I.)	*p*	OR (95% C.I.)	*p*
HDL-cholesterol	1.138 (0.996–1.301)	0.057	-	-
Atrial fibrillation	2.279 (1.054–4.931)	0.036	-	-
Mitral valve prolapse				
LVH	0.401 (0.186–0.867)	0.020	-	-
Mitral regurgitation				
CRP	1.005 (1.001–1.009)	0.019	1.005 (1.001–1.009)	0.025
Dyslipidemia—Triglycerides	1.793 (1.019–3.154)	0.043	1.760 (0.985–3.145)	0.056
HDL-cholesterol	0.895 (0.800–1.002)	0.055	-	-
Secondary ST-T changes	0.486 (0.259–0.910)	0.024	0.522 (0.275–0.989)	0.046
Pulmonary arterial hypertension				
Rheumatoid factor	1.008 (1.000–1.016)	0.063	-	-
HDL-cholesterol	1.113 (0.979–1.264)	0.102	-	-
Pulmonary regurgitation				
Left bundle branch block	7.264 (1.656–31.869)	0.009	-	-

**Table 10 medicina-60-00038-t010:** Comparison of analyzed parameters between patients according to the existence of atrial fibrillation (* Fisher’s Exact Test, ** Mann–Whitney *U* Test, *** Pearson Chi-square Test, **** Fisher’s Exact Test with Bonferroni corrected Z-tests).

Parameter/Group	Atrial Fibrillation		*p*
	Absent (*n* = 147)	Present (*n* = 53)	
Gender (Male) (Nr., %)	63 (42.9%)	13 (24.5%)	0.021 *
Age (Median (IQR))	63 (55–73)	61 (55–70.5)	0.642 **
Stages of RA disease (Nr., %)			0.890 *
Stage I	20 (13.6%)	6 (11.3%)	
Stage II	28 (19%)	10 (18.9%)	
Stage III	60 (40.8%)	25 (47.2%)	
Stage IV	39 (26.5%)	12 (22.6%)	
Smoking (Nr., %)	59 (40.1%)	11 (20.8%)	0.012 *
Alcohol consumption (Nr., %)	37 (25.2%)	10 (18.9%)	0.450 *
Hypertension (Nr., %)	85 (57.8%)	21 (39.6%)	0.025 *
Hypercholesterolemia (Nr., %)	72 (49%)	19 (35.8%)	0.110 *
Diabetes mellitus (Nr., %)	24 (16.3%)	4 (7.5%)	0.165 *
CRP (Median (IQR))	110 (2–135)	114 (3–147)	0.505 **
Fibrinogen (Median (IQR))	4.7 (4.5–5.1)	4.6 (4.5–4.85)	0.022 **
ESR (Median (IQR))	98 (14–106)	98 (15.5–110.5)	0.535 **
Rheumatoid factor (Median (IQR))	201 (121–213)	206 (168.5–213.5)	0.047 **
Atherogenic coefficient (Median (IQR))	3.375 (2.75–4.75)	2.90 (2.29–4.67)	0.016 **
Total cholesterol (Median (IQR))	180 (157–230)	160 (145–222.5)	0.003 **
Dyslipidemia—Total cholesterol (Nr., %)	72 (49%)	19 (35.8%)	0.110 *
Triglycerides (Median (IQR))	150 (100–170)	110 (85–157.5)	0.028 **
Dyslipidemia—Triglycerides (Nr., %)	74 (50.3%)	20 (37.7%)	0.148 *
HDL-cholesterol (Median (IQR))	40 (40–44)	40 (40–44.5)	0.460 **
Dyslipidemia—HDL-cholesterol (Nr., %)	24 (16.3%)	5 (9.4%)	0.262 *
LDL-cholesterol (Median (IQR))	116 (95–154)	100 (83.5–149)	0.008 **
Dyslipidemia—LDL-cholesterol (Nr., %)	65 (44.2%)	18 (34%)	0.255 *
Coronary artery disease (Nr., %)			0.030 ****
Absent	64 (43.5%)	33 (62.3%)	
Monovascular artery disease	46 (31.3%)	16 (30.2%)	
Bivascular artery disease	24 (16.3%)	3 (5.7%)	
Trivascular artery disease	13 (8.8%)	1 (1.9%)	
Pericarditis (Nr., %)	20 (13.6%)	14 (26.4%)	0.033 ***
Mitral valve prolapse (Nr., %)	33 (22.4%)	13 (24.5%)	0.849 *
Mitral regurgitation (Nr., %)	63 (42.9%)	24 (45.3%)	0.872 *
Pulmonary arterial hypertension (Nr., %)	24 (16.3%)	15 (28.3%)	0.070 *
Tricuspid regurgitation (Nr., %)	102 (69.4%)	37 (69.8%)	1.000 *
Pulmonary regurgitation (Nr., %)	90 (61.2%)	38 (71.7%)	0.186 *
CHF etiology			
Hypertension (Nr., %)	69 (46.9%)	20 (37.7%)	0.264 *
Coronary artery disease (Nr., %)	30 (20.4%)	5 (9.4%)	0.091 *
Valvular heart disease (Nr., %)	18 (12.2%)	10 (18.9%)	0.252 *
Arrhythmias (Nr., %)	1 (0.7%)	14 (26.4%)	<0.001 *

**Table 11 medicina-60-00038-t011:** Logistic regression models used for the prediction of atrial fibrillation (* multivariable model selected based on the Forward stepwise selection).

Parameter	Univariable		Multivariable *	
	OR (95% C.I.)	*p* *	OR (95% C.I.)	*p* *
Gender (Female)	2.308 (1.139–4.674)	0.020	-	-
Smoking	0.391 (0.186–0.820)	0.013	-	-
Hypertension	0.479 (0.252–0.908)	0.024	-	-
Fibrinogen	0.301 (0.107–0.845)	0.023	-	-
Rheumatoid factor	1.009 (1.001–1.017)	0.020	-	-
Atherogenic coefficient	0.720 (0.547–0.947)	0.019	-	-
Total cholesterol	0.990 (0.982–0.998)	0.011	0.991 (0.983–0.999)	0.034
Triglycerides	0.991 (0.983–0.999)	0.029	-	-
LDL-cholesterol	0.988 (0.979–0.997)	0.012	-	-
Coronary artery disease	0.467 (0.245–0.890)	0.021	0.507 (0.259–0.992)	0.047
Pericarditis	2.279 (1.054–4.931)	0.036	2.401 (1.078–5.350)	0.032

## Data Availability

The data presented in this study are available on request from the corresponding authors.

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
