# Peer review of "Prevalence of Cardiovascular Comorbidities in Patients with Rheumatoid Arthritis"

_medicina, 2023, doi:10.3390/medicina60010038_

Round 1

Reviewer 1 Report (Previous Reviewer 1)

Comments and Suggestions for Authors

The authors have made all relevant corrections, but the first paragraph of the Introduction needs to have the references properly inserted (e.g., the Danish study is not referenced). 

Comments on the Quality of English Language

Language is better. Please check for minor mistakes.

Author Response

Reviewer 2 Report (Previous Reviewer 3)

Comments and Suggestions for Authors

The authors have observed most of the recommendations of the reviewers and have significantly improved the quality of their manuscript. It can now be published in its current version.

Comments on the Quality of English Language

The English language is generally fine.

Author Response

Reviewer 3 Report (New Reviewer)

Comments and Suggestions for Authors

The four stages of RA? Which classification? clinical/ Radiological? please justify the bibliography 

the cardiovascular diseases - must be named in matherial and methods

the matherial and methods :study design, prospectivelly. transversal / interventional- when patients started the statins during the study - duration of drug administration follow up ?

no patient on biological  therapy or target therapy?  it is not clear if this medication are exclusion criteria, being known that usually the Rheumatologists used this kind of medication ?

Dislipidemya - the value of LDL higher than...?

                                             HDL lesser than...?

it is expectable that Disease activity (CDAI/ DAS 28CRP) must be comparable

the group must be more extensivelly described before starting the comparisons

Major revisions are required in order to be clear the article structure, and to easy follow up the informations

the bibliography is quite recent 

Comments on the Quality of English Language

none

Author Response

This manuscript is a resubmission of an earlier submission. The following is a list of the peer review reports and author responses from that submission.

Round 1

Reviewer 1 Report

Comments and Suggestions for Authors

Please see attached PDF for review comments.

Comments on the Quality of English Language

The writing itself seems OK at first but actually needs some editing work. 

Reviewer 2 Report

Comments and Suggestions for Authors

Attached my comments 

Comments on the Quality of English Language

A native English speaker to review the manuscript  is necessary 

Reviewer 3 Report

Comments and Suggestions for Authors

The authors of the manuscript present results from a retrospective study, including 200 patients with rheumatoid arthritis at different stages of the disease, in which they investigate the type and prevalence of cardiovascular complications. This article is topical and important  from a clinical and scientific point of view as rheumatoid arthritis is very common among the general population of many countries. This autoimmune disease may affect not only the joints but also other tissues and organs, including the heart structures. 

I have the following recommendations to the authors:

1. I suggest that the prefix "The" be omitted in the manuscript title.

2. The abstract must be shortened at least twice. Please, put some numbers/values/percentages and p-values at the parts of the abstract, presenting results from the study.

3. The introduction must be more succinct and better structured. The authors may start will a short description of the nature of rheumatoid arthritis - the etiology and pathogenesis (5-7 sentences at most as) and then focus on the cardiovascular involvement in this disease. The introduction is expected to be no longer than 1/2 to 2/3 of a standard page. Additional information could be presented in the Discussion. 

4. Better formulation of the inclusion and exclusion criteria is needed. The authors have stated the that "no patient has been excluded from the study" which sounds a little bit superficial and questionable for a scientific article. They may point out as exclusion criteria for example: 1. Age <18 years; 2. Cardiovascular comorbidities/complications in patients with rheumatoid arthritis, due to other concomitant conditions/diseases, etc.

5. What was the coronary angiography - CT or invasive? For what indication has it been conducted (chronic /acute coronary syndromes)?

6. The authors have stated in the beginning of "Materials and Methods" that their manuscript is based on a retrospective study. However, further in the same section it becomes clear they have performed a prospective follow-up for a certain period of time ("We monitored groups..."). Please, specify. 

7. There are missing p-values in some of the tables. Values in table 2  are presented only as numbers without percentage.

8. The first sentence of the "Results" is unnecessary. This information has already been shown in "Materials and Methods". 

9. The sentence in lines 209/210 is a type of discussion and should be moved to "Discussion". 

10. The study is missing a control group. On the other hand, the investigated concomitant risk factors and comorbidities are strong enough to cause by themselves various cardiovascular complications, including coronary artery diseases. So, the study design does not allow assessment of the impact of rheumatoid arthritis on the cardiovascular risk.

11. The conclusion is too large. It has to be just a few sentences. 

Comments on the Quality of English Language

English needs just a few corrections

Round 2

Reviewer 1 Report

Comments and Suggestions for Authors

Although some requested revisions were made, others were not. Please create a point-by-point response letter that indicates the corrections and justifies the decisions. This a traditional way to respond to reviews as it is customary to respond to EACH issue a reviewer raises.

Not making the requested changes and not listing any scientific/logical justification will not get the paper accepted. 

Comments on the Quality of English Language

The language is still poor. This paper will require extensive editing. 

Reviewer 2 Report

Comments and Suggestions for Authors

A letter  point by points response is not present. The diagnosis od rheumatoid arthritis is made without any criteria, the disease activity is not reported. Summary remains as in the first submission.mean value of crp, ear etc are not reported

Comments on the Quality of English Language

I think an review by native English speaker is necessary 

Reviewer 3 Report

Comments and Suggestions for Authors

The authors of the manuscript have observed most of the reviewer's recommendations and have corrected/added their work accordingly. The article has been significantly improved compared to its initial version. It can now be considered by the Editors of the journal for publication in its current form. 

Comments on the Quality of English Language

English language is generally fine, just a few stylistic corrections could be made (not mandatory however).
